# Dynamics of Quasi-One-Dimensional Structures under Roughening Transition Stimulated by External Irradiation

**DOI:** 10.3390/nano12091411

**Published:** 2022-04-20

**Authors:** Vyacheslav N. Gorshkov, Volodymyr V. Tereshchuk, Oleksii V. Bereznykov, Gernot K. Boiger, Arash S. Fallah

**Affiliations:** 1Igor Sikorsky Kyiv Polytechnic Institute, National Technical University of Ukraine, 37 Prospect Peremogy, 03056 Kiev, Ukraine; volodymyr379@gmail.com (V.V.T.); alexber36@gmail.com (O.V.B.); 2Department of Chemistry and Biomolecular Science, Clarkson University, Potsdam, NY 13699, USA; 3Institute of Computational Physics, Zürich University of Applied Sciences, Wildbachstrasse 21, 8401 Winterthur, Switzerland; boig@zhaw.ch; 4Department of Mechanical, Electronic and Chemical Engineering, OsloMet, Pilestredet 35, St. Olavs Plass, 0130 Oslo, Norway; arashsol@oslomet.no

**Keywords:** nanostructure instabilities, nanowire breakup, roughening transition, anisotropy of surface energy density, Monte Carlo kinetic model

## Abstract

We studied the striking effect of external irradiation of nanowires on the dynamics of their surface morphology at elevated temperatures that do not destroy their crystal lattice. Numerical experiments performed on the basis of the Monte Carlo model revealed new possibilities for controlled periodic modulation of the cross-section of quasi-one-dimensional nanostructures for opto- and nanoelectronic elements. These are related to the fact that external irradiation stimulates the surface diffusion of atoms. On the one hand, such stimulation should accelerate the development of the well-known spontaneous thermal instability of nanowires (Rayleigh instability), which leads to their disintegration into nanoclusters. On the other hand, this leads to the forced development of the well-known roughening transition (RT) effect. Under normal circumstances, this manifests itself on selected crystal faces at a temperature above the critical one. The artificial stimulation of this effect on the lateral surface of quasi-one-dimensional structures determines many unpredictable scenarios of their surface dynamics, which essentially depend on the orientation of the nanowire axis relative to its internal crystal structure. In particular, the long-wave Rayleigh breakup observed in absence of external irradiation transforms into strongly pronounced short-wave metastable modulations of the cross-section (a chain of unduloids). The effect of the self-consistent relationship between the Rayleigh instability and RT is dimensional and can be observed only at relatively small nanowire radii. The fact is analyzed that, for the manifestation of this effect, it is very important to prevent significant heating of the nanowire when surface diffusion is stimulated. A number of developed theoretical concepts have already found confirmation in real experiments with Au and Ag nanowires irradiated by electrons and Ag^+^ ions, respectively.

## 1. Introduction

The physical and chemical properties of semiconductor and metallic nanowires are of growing interest, due to their potential applications in a number of electronic, optoelectronic, and electromechanical devices. In particular, the advantageous characteristics of face-centered cubic (FCC) nanowires allow the use of these structures in biosensors [1,2,3,4], elements of solar panels [5,6], as well as in plasmonic waveguides [7]. High electrical and low thermal conductivity [8,9], optical absorption characteristics [10,11,12], and quantum confinement effects [13] make silicon nanorods potentially useful as building blocks for thermoelectric and photonic devices such as nanoconductive field-effect transistors [14,15,16], hypersensitive biological, chemical or mass sensors [17,18,19,20,21,22,23,24,25], and photodetectors [26,27,28]. The electrical and optical properties of quasi-one-dimensional nanostructures essentially depend on the morphology of their surface, which may vary periodically along the axis. In particular, modern methods of nanostructure synthesis make it possible to create 1D core–shell nanowires with different surface morphology that are now used in solar energy conversion and electrochemical energy storage [29,30,31]. In our study, we consider the theoretical foundations of methods for creating periodically modulated structures based on the general physical concepts expounded in the sequel.

The transformation of a nanowire initially uniform in cross-section into an ordered chain of nanobeads is associated with mass transfer along its surface. This transfer is due to the surface diffusion of atoms, which increases with increasing temperature. Spontaneous periodic modulation of the surface must satisfy a number of thermodynamic relations and equations of motion of the medium. The first requirement allows a large set of “trajectories” for the dynamic evolution of the system, which is subsequently significantly narrowed down when it is necessary to satisfy the equations of motion. The selected trajectories lead to the dominance of periodic modulations of the nanowire radius with a wavelength, $\lambda_{max}$, which corresponds to the maximum growth increment, $\gamma_{max}$. The value of $\lambda_{max}$ is determined by the inhomogeneity of the distribution over the surface of sets of two probabilities. The first characterizes the probability of atoms jumping, which depends on the energy of the activation barrier in the initial state. If the jump occurs, then the probabilities of its directions depend on the number of nearest neighbors in the new positions. Different relations between the distributions of the two indicated types of probability, which depend on temperature, lead to corresponding different wavelengths of perturbations. The frequently observed breakup of nanowires into individual nanodroplets is often associated with the Rayleigh instability of liquid jets. However, the approach of isotropic surface energy density used in this particular case significantly reduces the applicability of the developed physical concepts for interpreting the dynamics of crystalline nanostructures. For nanowires with pronounced anisotropy, the “number of degrees of freedom” increases and determines a greater variety of scenarios for the nanowire surface dynamics. By varying its temperature and axis orientation, the relationship between the two sets of probabilities mentioned above changes. The additional degree of freedom associated with an artificial change in the probability distribution of jumps leads to even greater diversity in the evolution of nanostructures. The purpose of our research is shown in the context of a brief review of the previous results by predecessors.

The disintegration of nanowires into separate fragments as a result of thermal instability has been studied in numerous experimental [32,33,34,35,36,37,38,39] and theoretical studies [40,41,42,43,44,45]. In the Nichols and Mullins model [40], it was assumed that the surface energy density, σ, of a nanowire is isotropic (does not depend on the orientation of a given fragment of the lateral surface with respect to the preserved internal crystal structure of the nanowire). In the absence of internal flows, the surface diffusion of atoms determines the dynamics of the nanowire surface morphology. However, the result obtained for the wavelength of perturbations, $\lambda_{max}$, growing in time with a maximum increment, $\gamma_{max}$, coincides with the result of the classical Rayleigh theory: $\lambda_{max}=9.02r_{nw}$ where $r_{nw}$ is the initial radius of the nanowire [40,46]. The disintegration of nanowires into nanoclusters, the volumes of which correspond to the volume of the section of the initial nanowire with a length $\lambda_{max}$, was indeed observed in experiments [38,39]. However, in numerous studies, significant deviations from the predictions of the Nichols and Mullins theory ($\lambda_{max}/r_{nw}∼25-30$ в [33,34,36]) are noted. The reasons for these deviations are related to the anisotropy of the surface energy density and were studied in detail in recent research [46,47,48]. It is shown that, depending on the orientation of the nanowire axis relative to its internal crystal structure, the value of the breakup parameter β
$\left(\beta =\lambda_{max}/r_{nw}\right)$) can be either noticeably higher than 9 or lower than the threshold value 2π (in the case of isotropic σ, the nanowire surface is stable against perturbations of the radius with wavelength $\lambda <\lambda_{cr}=2\pi r_{nw}$).

The next factor that can be responsible for nanowire transformation is the roughening transition (RT) [49,50,51,52,53,54,55,56,57]. If the temperature of the system is greater than a certain critical value ($T>T_{R}$), then periodic “hillocks and cavities” develop on a smooth crystalline face. It is worth noting that this effect occurs only on selected crystal faces. Its occurrence on the side surface of the nanowire is not ruled out either, since, at the initial stage of transformation of a cylindrical nanowire, it is fragmentarily limited by longitudinal edge strips with the lowest possible total surface energy (Figure 1A(a)). Thus, the dynamics of the surface of a nanowire can be determined by two physical mechanisms—Rayleigh instability and RT—which arise with an increase in temperature and are associated with the intensification of the surface diffusion of atoms. However, we are not aware of publications in which the authors of experimental studies discuss any manifestation of RT in the breakup of nanowires when interpreting the results obtained. One of the probable reasons is that the temperature of the nanowire during a breakup is below the critical one, $T<T_{R}$. It is also possible that the Rayleigh instability development increment is much higher than the RT increment.

In our research, we study the possibilities of controlling the surface dynamics of quasi-one-dimensional nanostructures in the case of their surface exposure to external irradiation (for example, with light, an external electron flow or $Ar^{+}$-ions of a cold plasma), which initiates the manifestation of RT. In this case, it is important that the external action, which stimulates the surface diffusion of atoms, does not lead to significant heating of the wire. It sounds paradoxical, but a possible “overheating” may eliminate the role of RT against the background of the development of the Rayleigh instability. A weighty argument for our theoretical studies was the results of experiments with gold [32] (electron irradiation) and silver [58] nanowires (irradiation with $Ar^{+}$-ions). The most interesting scenarios for the transformation of nanowires arise at small radii—$r_{nw}≲5\;nm$.

The requirements of smallness for the cross-section of a nanowire are clearly demonstrated in Figure 1, which shows the results of our numerical simulations. Surface modulations appear on the (100)-type facet strips precisely because of the stimulated RT. One can see that the length of the formed nanohillocks is comparable to the radius $r_{nw}$$\left(l_{x}\sim r_{nw}\right)$), which is noticeably lower than the so-called “energy threshold”, which prohibits the fragmentation of the nanowire into fragments with a length, l, where $l<∆l_{min}\approx 4.5r_{nw}$ (at $l=∆l_{min}$, the surface energy of the formed nanodroplets is equal to the surface energy of the initial nanowire). Thus, the RT effect on the surface of a relatively thick nanowire will only lead to small-scale perturbations of the surface without a significant self-consistent phenomenon with the influence of Rayleigh instability on the subsequent dynamics of its morphology. In a rough approximation, such an influence should be expected in the case of $l_{x}≳4.5r_{nw}$, or $r_{nw}≲l_{x}/4.5\approx 12a$. Thus, for gold, the stimulated RT effect should most strikingly appear at $r_{nw}≲5\mathrm{nm}$. This estimate is in good agreement with the results of numerical simulations obtained below.

In the present study, we show that stimulation of the surface diffusion of atoms by external irradiation is an effective tool for controlling the parameters of generated quasi-one-dimensional periodic nanostructures. Depending on the orientation of the nanowire axis, the breakup parameter, β, can be controlled over a wide range. For instance, with the same initial radius of the (110)-oriented nanowire, the value of β can be reduced from β~25−30 (disintegration into “nanodroplets” in the absence of irradiation) to β~4−5 (formation of unduloids [59]—frozen in time-periodic modulations of the nanowire cross-section). An inverse dependence of the parameter *β* on the level of stimulation of surface diffusion is also possible: in the case of the (111) orientation, an increase in the irradiation intensity transforms relatively short-wavelength perturbations (β<9) into long-wavelength ones (β~20). Therefore, through fragmentary irradiation of a nanowire, it is possible to generate chains of nanoclusters from a single nanowire with a variable frequency of gaps between them. These have potential applications in biomedicine and spectroscopy [60]. In this research, the influence of the stimulated roughening transition on the evolution of nanowires is studied, with a detailed qualitative analysis of the ongoing physical processes based on the Monte Carlo model—as presented in the next section.

## 2. Numerical Model

The applied Monte Carlo (MC) model has been successfully used to simulate nonequilibrium surface growth [61], synthesis of nanoparticles [62], and formation of nanopillars and nanoclusters for sintering and catalysis applications [63,64,65]. The present numerical approach has also been applied to study the disintegration of nanowires with diamond-like and body-centered cubic (BCC) lattice structures [49,50]. Based on this MC numerical model, the synthesis of periodical core–shell nanostructures was also investigated [66]. In this section, we only present the basic concepts of the given MC method. A complete description of the applied MC approach can be found in [66].

In the simulations conducted, the computational domain has a cylindrical shape of length, L and radius R. The nanowire of radius $r_{nw}$
$\left(R\gg r_{nw}\right)$), with its atoms occupying the lattice sites of an FCC lattice structure, is inserted into the domain in such a way that the axes of the nanowire and the cylinder are coincident. Surface atoms can hop to the nearest vacancies or detach from the nanostructure and become free according to probabilistic relations given below. The detached atoms hop in random directions within the boundaries of the computational domain with a constant length step, l, the magnitude of which equals half of the lattice constant, a. The walls of the container are assumed reflective, so the number of atoms in the entire system under consideration is time-invariant. Free atoms can reattach to the surface of the crystalline structure at vacant sites that are adjacent to the sites occupied by nanostructure atoms. Each site is represented by a region of space within the Wigner–Seitz unit cell in the lattice. When an atom jumps into one of Wigner–Seitz’s cells, it is located at the center of that cell and becomes attached to the nanostructure.

To simulate the surface dynamics of nanostructures, two parameters are used in the model. The first parameter α has its magnitude as ε scaled per kT, where k is the Boltzmann constant, T is the temperature of the system, and ε<0  reflects the local binding of adjacent atoms:(1)$$\alpha =\left|\varepsilon \right|/kT$$

The second parameter, p, determines the surface diffusion coefficient and depends on the activation/energy barrier, Δ, defined as follows:(2)$$p=\exp\left(-\Delta /kT\right)$$

For a system consisting of $N_{0}$ atoms one MC step consists of the following operations repeated $N_{0}$ times: (1) the random selection of an atom; (2) the determination of its new position. Thus, on average, each atom is selected once per one MC step. If an atom has $n_{vac}$ unoccupied neighboring lattice sites, it may jump into one of them with the probability $p_{jump}$ calculated as follows:(3)$$p_{jump}=p^{m_{0}}$$

where $m_{0}$ represents the number of nearest neighbors (occupied lattice sites).

If the jump occurs, the new position of the atom is chosen from the $n_{vac}+1$ lattice sites (the initial state is also included). The “direction” of the jump made is randomly chosen according to the probabilities of the target sites, $p_{target}^{\left(i\right)}$$(i=1,2,3,\ldots n_{vac}+1),$ which are equal to
(4)$$p_{target}^{\left(i\right)}=\gamma \times \exp\left(\frac{m_{t}^{\left(i\right)}\left|\varepsilon \right|}{kT}\right)\;𝛾={\left[\sum_{j=1}^{n_{vac}+1}\exp\left(m_{t}^{\left(j\right)}\left|\varepsilon \right|/kT\right)\right]}^{-1}$$

where $m_{t}^{\left(i\right)}$ denotes the number of nearest neighboring atoms in the assumed new state, indexed by (i). Therefore, the final position is determined by a set of Boltzmann factors.

In our model, a change in the temperature is reflected by varying the parameter α (note that α~1/T , using Equation (1)). The relation between the parameters α and p is described by the following equation [48]:(5)$$p={\left(p_{0}\right)}^{\alpha /\alpha_{0}}$$

The good agreement between experimental results and our previous numerical studies [46,47,48] was obtained using the following reference values for the FCC lattice: $\alpha_{0}=1.0$ and $p_{0}=0.7$. In subsequent numerical simulations, the effect of external irradiation on the surface diffusion of atoms is included in the increase in the parameter p, which is responsible for the jump frequency.

Let us expound on some technical details of the applied model. When modeling the breakup of nanowires, we make the last five atomic layers on both ends of the nanowire composed of frozen (“motionless”) lattice atoms. This is an undertaking equivalent to the imposition of periodic boundary conditions and is conducted to exclude the so-called end effects [47,48] at the early stages of the disintegration process. Our MC model considers only the interaction of atoms with their nearest neighbors. We do not consider the interaction of atoms with second- or higher-order neighbors, mainly because according to experimental studies [67] these make a small contribution to the potential or force in the case of an FCC lattice. Additionally, when modeling the dynamics of an unstable system, we do not specify the type of initial perturbations as they develop naturally due to thermal fluctuations inherent in this model.

Before presenting the results obtained, the following remark is in order: The RT effect on a flat crystalline face is associated with the formation of clusters from atoms that have jumped from the same face into the near-surface layer of vacancies (we mark this layer with the index +1). An increase in the crystal temperature leads to a decrease in the degree of anisotropy both in the frequency of hops over the surface of the structure (the parameter *p* increases) and the probability distribution over the possible directions of hops (the parameter *α* decreases). In this case, two physical processes compete. On the one hand, the flow of atoms from the surface 0-layer to the +1-layer increases, increasing the density of mobile atoms in this layer. On the other hand, an increase in temperature hinders homogeneous nucleation in a two-dimensional/surface “gas” of these atoms. The dominance of the first factor is achieved at *T > T**_k_* only on selected crystalline faces and determines the appearance of ordered structures on them with a characteristic size, *l_RT_*. Given the possibility of developing such an effect on the lateral surface of the nanowire, there is no reason to compare the size of the fragments during its breakup with the classical parameter 9*r_nw_*.

The natural manifestation of RT in the dynamics of one-dimensional nanosystems was considered in our studies for BCC [50] and diamond-like crystal lattice structures [49]. In the latter case, the RT effect resembles the case of unstable diffusion growth that is self-consistent with its own vapor. The variant of stimulated mobility of surface atoms studied in this research opens up possibilities for the controlled creation of a wider class of one-dimensional structures modulated over the cross-section. In this case, it is possible to realize a kind of interference of surface perturbations with a wavelength $\lambda_{max}$, which arise due to the thermal instability of the nanowire in the absence of external irradiation, and additionally stimulated perturbations with a characteristic size $l_{RT}$. Naturally, the most interesting scenarios of the nanowire dynamics correspond to the cases where $l_{RT}≲\lambda_{max}$.

Notably, for the chosen orientations of the nanowire, the RT may not develop on its lateral surface. In this case, its dynamics will correspond to the manifestation of thermal instability with a modified parameter p.

## 3. Results

### 3.1. Physical Mechanisms of Stimulated Roughening Transition

As we have already noted, the initial surface of the nanowire transforms with minimization of the total surface energy at elevated temperatures and can be represented by a set of facet strips of various types (one of the possible variants of such an intermediate stage is shown in Figure 1A(a)). Further dynamics of its surface morphology, as will be shown below, is associated with a complex system of surface flows of atoms along the “strips” and with the exchange of atoms between adjacent strips. Now we demonstrate the manifestation of stimulated RT on isolated facet strips of various types (Figure 2). We consider the dynamics of atoms located in the near-surface layers of an extended facet. It is assumed that only atoms filling the plate with dimensions L(length), w (width), and h (height) are mobile. The movable atoms cannot come out of the perimeter of the upper facet. The object chosen for numerical modeling clearly demonstrates the main results of stimulation of the surface diffusion of atoms. The unit of length in our model is the distance between atomic layers of the (100) type in the FCC crystal lattice. In the case of gold, this unit of length corresponds to approximately 2Å.

In our model, the effect of external irradiation on the hopping frequency of surface atoms corresponds to choosing a value of the parameter p for a given α, which is slightly higher than the value given by relation (5). At the initial stage, the near-surface atomic layer (+1-layer) is filled with atoms of the initial 0-layer, and in the process of diffusion over this (initial) layer small clusters can form (Figure 1A(a)). However, the formed clusters are unable to generate second-generation clusters in the +2-layer. Ordered pronounced modulations of the plate surface arise only when it is irradiated, which increases the frequency of atomic jumps from the i-th layer to the $\left(i+1\right)$-th layer. Furthermore, the intensification of the surface diffusion of atoms on this layer reduces the time of formation of new clusters with supercritical sizes relative to their decay. Naturally, this process occurs faster on the (111) face (see Appendix A), since in the formed (111) clusters there are more interatomic bonds than in (100) clusters. Additionally, they are more stable at small sizes, which is essential at the initial stage of nucleation. Notably, we did not observe stimulated RT on the (110) facet for the parameters chosen above. This result is associated with the following factors: (1) the low surface density of atoms on the (110) face (the low binding energy between atoms to create a new nucleus in +1-layer) and (2) the low mobility of single atoms in this layer.

For the observed modulations in Figure 2, the surface energy of the plate increases, since the area of the modulated surface exceeds the initial area. However, such a process does not contradict the laws of thermodynamics. Our model assumes that the temperature of the system is constant, and in this case, all processes accompanying a decrease in the Helmholtz free energy, F: dF=dU−TdS, are allowed. In the case under consideration, the increase in surface energy, U, is offset by an increase in entropy associated with an increase in the number of surface states. However, general thermodynamic concepts do not reflect the details of the kinetics of the process and do not answer the question of what physical mechanisms lead to the formation of hillocks and cavities on the facet strips. A more intricate picture arises in the dynamics of surface flows of nanowires and, accordingly, in the dynamics of their surface morphology.

Therefore, we show in the sequel how changes in the model parameters, p and $\alpha =\left|\varepsilon \right|/kT$, change the course of the process in the most unexpected way.

Let us make some estimates for the surface fluxes on the lateral surface of the wire with developed radius modulations and on the lateral surface of the formed hillock (Figure 3). For Figure 3A, it is assumed that the sizes of atomic layers differ little from each other, and when a contour (peripheral) atom jumps from layer i, it becomes a contour atom in one of the nearby layers. Then, the flow of atoms from layer i to layer i+1, $\Phi^{+}$, and the reverse flow, $\Phi^{-}$, are equal to
(6)$$\Phi^{+}=\Gamma_{i}\times p^{\langle n_{bb}\rangle {}_{i}}\times \left\{\gamma_{i}\exp\left[\langle n_{bb}\rangle {}_{i+1}\alpha \right]\right\}\;\Phi^{-}=\Gamma_{i+1}\times p^{\langle n_{bb}\rangle {}_{i+1}}\times \left\{\gamma_{i+1}\exp\left[\langle n_{bb}\rangle {}_{i}\alpha \right]\right\}$$

Here, Γ is the number of contour atoms at the layer boundary and $\langle n_{bb}\rangle$ is the average number of bonds for the contour atoms of the layer. The values in curly brackets represent the Boltzmann factor (4) for jumping atoms from layer to layer:(7)$$\gamma_{i}={\left(\exp\left[\langle n_{bb}\rangle {}_{i-1}\alpha \right]+\exp\left[\langle n_{bb}\rangle {}_{i}\alpha \right]+\exp\left[\langle n_{bb}\rangle {}_{i+1}\alpha \right]\right)}^{-1}\;\gamma_{i+1}={\left(\exp\left[\langle n_{bb}\rangle {}_{i}\alpha \right]+\exp\left[\langle n_{bb}\rangle {}_{i+1}\alpha \right]+\exp\left[\langle n_{bb}\rangle {}_{i+2}\alpha \right]\right)}^{-1}\;\langle n_{bb}\rangle {}_{i-1}>\langle n_{bb}\rangle {}_{i}>\langle n_{bb}\rangle {}_{i+1}>\langle n_{bb}\rangle {}_{i+2}$$

As the parameter p increases, the incremental left-to-right flow change, ${\Delta \Phi }^{+}$, exceeds the incremental right-to-left flow change, ${\Delta \Phi }^{-}$, at least under the condition that $d\left(p^{\langle n_{bb}\rangle {}_{i}}\right)>d\left(p^{\langle n_{bb}\rangle {}_{i+1}}\right)$, or
(8)$$\frac{\langle n_{bb}\rangle {}_{i}}{\langle n_{bb}\rangle {}_{i+1}}p^{\langle n_{bb}\rangle {}_{i}-\langle n_{bb}\rangle {}_{i+1}}>1$$

The threshold value of the parameter p for stimulated RT (creation of conditions for the flow of atoms from the wide to the narrow part of the nanostructure) may be roughly estimated from the following considerations. The maximum average number of horizontal bonds at the boundary of a (111) cluster with a hexagonal structure is about four, and the minimum is about three. Then, in accordance with inequality (8), stimulation of surface diffusion by irradiation should provide the parameter p around the following value:(9)p≳0.75

The obtained rough estimate agrees with the results of our numerical simulations.

Results in Figure 4 demonstrate the possibility of a sharp change in the nature of the dynamics of a dumbbell-shaped nanocluster by variation in the parameter p. In the absence of irradiation, this structure breaks up into two parts, since the surface diffusion fluxes are directed away from the area of its initial necking. Stimulation of surface diffusion (an increase in the parameter p to 0.825) changes the direction of diffusion fluxes in the central part and leads to the formation of an integral cluster. The observed dramatic change in the scenario of system evolution is not due to an increase in the frequency of hops and acceleration of mass transfer but owing to the fact that an increase in p reduces the degree of inhomogeneity in the distribution of the frequency of hops of bound atoms over the surface of the nanostructure.

It is quite difficult to estimate the role of system heating during irradiation. The point is that, according to relations (6) and (7), any additional heating that accompanies an increase in the parameter p leads to the following inequality:(10)$$d\left(\gamma_{i}\exp\left[\langle n_{bb}\rangle {}_{i+1}\alpha \right]\right)>d\left(\gamma_{i+1}\exp\left[\langle n_{bb}\rangle {}_{i}\alpha \right]\right)$$

This means a higher flow rate from left to right than in the reverse direction, and this is indeed observed in our numerical simulations, in which heating a nanowire significantly changes the transformation of its surface (see below). However, for nanohillocks, the conclusion made cannot be applied. The reason is that the chains of boundary atoms of layers i and i+1 are separated from each other (Figure 1, Figure 2 and Figure 3B). As a result, each atomic layer is represented both by an integral cluster and by the two-dimensional space surrounding it, into which the “evaporation” of cluster atoms can occur.

Quantitative analysis of the dynamics of such a system is difficult, but its end result is quite understandable. Heating during irradiation intensifies the destruction of clusters and inhibits the development of stimulated RT (Figure 5). Naturally, the destructive effect of heating is more pronounced on the (100) face with a smaller number of horizontal bonds in the cluster (four), compared to six horizontal bonds in the cluster on the (111) face. As a result, only strongly rarefied clusters are formed on the (100) face and only in two near-surface atomic layers.

### 3.2. Two-Mode Transformation of Nanowire Surfaces

The characteristic length of hillocks, $l_{hl}$, seen in Figure 1, Figure 2, and Figure 5, is about 100 units (50 lattice constants), which corresponds to 20 nm for Au. The $l_{hl}$ value is not related to the dimensions of the plate, as can be seen in Figure 2B, configuration (d). If the radius of the nanowire is chosen such that the length of modulations of its surface, λ, with the development of “Rayleigh” instability in the absence of irradiation is comparable to the length $l_{hl}$, upon stimulation of surface diffusion, a pronounced two-mode regime of the evolution of its surface will arise. The manifestation of this regime depends on the orientation of the nanowire since it determines the type of bands of its lateral surface at the initial stage (configuration (a) in Figure 1A). It is a two-mode regime of nanowire dynamics that we associate with the unexpected results recorded in experiments with gold nanowires [32]. During the first 10 min, pronounced modulations of the wire cross-section with a diameter of ~6 nm appeared and subsequently remained frozen in time for at least 30 min. The paper does not indicate the orientation of the nanowire axis, but we have a reason to believe that these results were obtained for the (100)-type orientation. In this case, at the initial stage of development of the “Rayleigh” instability, radius modulations with a wavelength of $\lambda \approx 5.5r_{nw}$ appear according to our numerical simulations [46] (a noticeable difference from the predictions of a theoretical study presented in [40]—the maximum instability increment corresponds to the length of perturbations $\lambda_{max}\approx 9r_{nw}$—is associated with a pronounced anisotropy of the surface energy density, σ [46].) Just the same ratio, $\lambda /r_{nw}\approx 5.5$, can be obtained based on data analysis [32]. At nanowire radius $r_{nw}\sim 3nm$, the length of “Rayleigh” instability perturbations, λ≈17nm, is close to the characteristic length $l_{hl}$. The above estimates indicate that in [32] it was the two-mode regime that was implemented, the characteristics of which are presented in detail in Figure 6.

In the irradiation mode, the modulations of the nanowire cross-section (Figure 6A, curve 2) saturate at the same time that is required to establish a quasi-stationary state on the (100)-type facet strip (see the lower dependency for $N_{bb}\left(t\right)$ in Figure 5B). This means that at this stage of the wire dynamics, RT dominates on the four (100)-facet strips of the lateral surface (configuration (a) in Figure 1A). The formation of hillocks is associated with the longitudinal transport of atoms along the (100) bands and with the transverse drift from adjacent (110) bands, on which RT does not develop. The development of constrictions increases the difference in the values of contour atoms, Γ, for neighboring atomic layers transverse to the axis of the nanowire, and leads to the fact that the flow of atoms to the narrowing region (Equation (6) for $\Phi^{+}$) almost compensates for the reverse flow (Equation (6) for $\Phi^{-}$). The further evolution of the nanowire surface is determined by the slow development of the Rayleigh instability. Visually, this evolution is almost imperceptible (Figure 6C). Similar to the results of the experiment [32], the transition time from configuration (a) to configuration (d) is three times longer than the time of formation of the configuration (a) itself; however, the changes in the shape of the nanowire are vanishingly small even in dependencies of the number of atoms in the atomic layers along nanowire axis (configurations (e) and (f) in Figure 6C).

If the wire is also heated up during irradiation, the effect of stimulated RT may not manifest itself in its dynamics (curve 3 in Figure 6A). One can see that the growth rate of nanowire modulations, $\delta \left(t\right)$, sharply decreases in the initial stage. The reason for the slow development of bottlenecks during heating was revealed by us when discussing relation (10). In addition, we recall that at the stage of developed modulations, stimulated surface diffusion also slows down their further development. These two factors lead to the fact that at the nonlinear stage of nanowire breakup, when neighboring “beads” can merge, the number of individual nanoclusters, $n_{cl}$, steadily decreases (which was confirmed in a series of analogical numerical experiments) in the sequence of regimes 1-3-2 represented in Figure 6A:(11)$$n_{cl}\left(1\right)>n_{cl}\left(3\right)>n_{cl}\left(2\right)$$
which is shown in configurations in Figure 6D(a’–c’).

### 3.3. Single-Mode Transformation of the Surface of Nanowires

In the absence of external irradiation, (110)-oriented nanowires are significantly resistant to thermal instability since, at the initial stage of transformation, they are limited by two pairs of (111)-type band facets with a low surface energy density (with a small number of broken bonds)—see Figure 7A. The wavelength with the maximal growth increment, $\lambda_{max}\sim \left(25-30\right)r_{nw}$ [42,46], is much higher than the predictions of Nichols and Mullins theory ($\lambda_{max}\sim 9r_{nw}$) [40]. For this orientation of the nanowire, cases become real when the length $\lambda_{max}$ noticeably exceeds the characteristic size of hillocks, $l_{hl}$. Then, it is the stimulated RT that plays a key role in the transformation of the nanowire surface. For Au, the relation $\lambda_{max}\approx 25r_{nw}>l_{hl}\approx 100$ is fulfilled at $r_{nw}≳1\;\mathrm{nm}$, meaning that when nanowires with a diameter of about $d_{nw}≳5\;\mathrm{nm}$ are irradiated, the dynamics of their surface evolve in a regime that we can physically reasonably call single-mode. However, in order for the effects of modification of the nanowire cross-section to be visually noticeable, its radius should not significantly exceed the height of these hillocks, $h_{hl}$ (according to our data, the height $h_{hl}\;$ reaches about 10 interatomic distances between adjacent (111) layers). The distance between (111) atomic layers is $2/\sqrt{3}$, and $h_{hl}\approx 12$. Therefore, the surface modulations become weakly pronounced at the nanowire radius $r_{nw}≳50$ ($r_{nw}≳10\;\mathrm{nm}$, in the case of Au).

The results of modeling single-mode decay are presented in Figure 7 and Figure 8. Note the weak dependence of the perturbation growth rate, $\delta \left(t\right)$, on the nanowire’s radius. There are also no noticeable differences in the wavelengths of surface perturbations, for which the ratio $\lambda_{max}/r_{nw}$ does not exceed 5.6. These two facts demonstrate the dominance of stimulated RT in the evolution of the nanowire. At relatively small radii, the development of hillocks (the number of broken bonds grows rapidly—Figure 7B) quickly leads to the rupture of the nanowire at the time of $\tau_{br}≲3\times 10^{6}$ MC steps. In comparison, in the absence of irradiation, a thinner wire ($r_{nw}=10.5$) experiences the first rupture at $t>12\times 10^{6}$ [46]. Naturally, metastable states with a pronounced modulation of the nanowire surface arise at radii $r_{nw}≳2h_{hl}$ (curves 3 in Figure 7A,B). Figure 8C demonstrates such a metastable state for a gold nanowire with a diameter of $d_{nw}≃8nm$. The main part of the nanowire (Figure 8C(e), $t=15\times 10^{6}$ MC steps) visually looks the same as at the time of its formation (Figure 8C(a), $t=2.25\times 10^{6}$ MC steps).

As the radius increases, the modulation value of the transverse dimensions of the nanowire becomes pronouncedly anisotropic (Figure 9(d,d’)). Even though at the early stages the lateral surface is formed by two pairs of (111)-type strips, as well as by two (100-type) strips (Figure 7A, inset), stimulated RT dominates on (111) facets, causing asymmetry in the deformations of the surface of the nanowire, which is visible in configuration (e) (Figure 9).

Thus, the physical mechanisms of surface morphology dynamics considered above indicate that, with a strictly theoretical consideration of the nanowire instabilities under irradiation, it makes no sense to look for the dependence of the characteristics of this instability on the parameter $\beta =\lambda /r_{nw}$, as was carried out when studying the Rayleigh instability.

### 3.4. Mode of Long-Wave Transformations

When the wire is oriented in the (111) direction, its lateral surface at the initial stage has the form of a hexagonal prism with lateral faces of the (110) type. The thermal instability of nanowires with this orientation, occurring without stimulation of surface diffusion, is characterized by the length of surface perturbations at the linear stage of about $\lambda \approx 7r_{nw}$ (Figure 10A). At $r_{nw}\approx 14$, the wavelength is λ≈100, which coincides with the length of the hillocks, $l_{hl}$, the effect of which on the wire dynamics was discussed above. However, the supposed two-mode transformation of the wire cannot be realized in this case. The fact is that, as we noted above, the RT effect does not develop on the (110) facet strips that bound the nanowire. However, stimulation of surface diffusion leads to a pronounced effect, registered in the experimental work of [58], in which silver nanowires were irradiated with cold argon plasma without significant heating. As a result, the break-up parameter $\beta =\lambda /r_{nw}$ increased significantly, compared with that observed in the absence of irradiation. This effect was also found in our numerical simulations (Figure 10B). The breakup parameter $\beta =\lambda /r_{nw}$ increases from the value of β≈7 (Figure 10A(d)) to the value of β≈18, which is in good agreement with the data presented in [58].

The physical mechanisms of the effect of a significant elongation of excited perturbations can be qualitatively explained based on the results obtained in the discussion of the flux balance in the bottleneck region (Equation (6), and Figure 3 and Figure 4). We demonstrated that, by stimulating surface diffusion, it is possible to prevent the breakup of a dumbbell nanocluster into two parts by changing the direction of atomic fluxes in the region of the previously formed bottleneck. In this case, the intensification of surface diffusion suppresses the almost simultaneous formation of neighbor bottlenecks at short distances (Figure 10A(a)) due to the intense diffusive mixing of atoms entering the near-surface layers when trying to form nanowire constriction regions. The result of such mixing is weakened only with an increase in the gap between the bottlenecks formed.

## 4. Discussion and Conclusions

Let us first expound on the theoretical aspects of the study conducted here on the dynamics of nanowire surfaces. Spontaneous transformation of a nanowire occurs with an increase in temperature, which increases the mobility of surface atoms. However, the scenario of this transformation is determined by both the degree of inhomogeneity in the frequency of jumps of bound atoms on the surface of the system and the degree of anisotropy in the probability of jumps in different directions (Equation (4)) into neighboring vacancies of the crystal lattice. These inhomogeneities are controlled by parameters *p* and α in our numerical model. Naturally, simple heating of the nanowire should be reflected both in an increase in the parameter p and a decrease in  α. The arising surface perturbation is observed in many experiments and is known as the Rayleigh instability. The scenarios of this instability often demonstrate significant deviations of the breakup parameter from the predictions of the classical theory $\beta =\lambda /r_{nw}\approx 9$, which is associated with the anisotropy of the surface energy density. Despite the fact that the indicated anisotropy introduces significant corrections into the mechanisms of nanowire surface dynamics, the developing instabilities have a common feature, which gives reason to still categorize them as one class of instability under the general term “Rayleigh instability”. Such a feature is scaling—the proportionality of the wavelengths of surface perturbations to the radius of the nanowire. In this formulation of the problem, the results of our previous numerical studies are in very good agreement with the experimental data.

In the present study, instabilities of a different type, for which the scaling condition is not satisfied, were studied. Such an instability regime is caused by the disruption of the “natural” relationship between the parameters p and α ($p={\left(p_{0}\right)}^{\alpha /\alpha_{0}}$, Equation (5)), which is determined by the temperature of the system. The stimulation of surface diffusion by an external action (irradiation) distorts this relationship ($p>{\left(p_{0}\right)}^{\alpha /\alpha_{0}}$) and increases the mobility of strongly bonded atoms sufficiently, which eventually leads to the observed instabilities. A return to the aforementioned natural relation between p and α, which can be achieved provided that the irradiation heats the system significantly and eliminates the development of a new type of instabilities.

The stimulation of the surface diffusion of atoms introduces various and significant corrections into the scenarios for the evolution of the nanowire surface morphology, which depends on the orientation of the nanowire relative to its internal crystal structure. If it is possible to develop an artificial RT effect on some facet strips of the nanowire lateral surface, then its result depends on the size ratio of the emerging hillocks, $l_{hl}$, with a characteristic length, λ, of the Rayleigh instability that could develop in the absence of irradiation (Section 3.2 and Section 3.3). With the chosen orientation of the nanowire axis, stimulation of surface diffusion might not lead to the RT excitation. In this case, the surface instability, which would arise even without external irradiation, expectedly increases the wavelength of perturbations under new physical conditions. The theoretical concepts developed in this study have been confirmed in several experiments [32,58].

The results obtained can be useful in flexible electronics, where ordered chains of Au and Ag nanoparticles, as well as radially modulated quasi-one-dimensional structures, are widely used (see Appendix A). In addition, the stimulation of surface diffusion can find applications in the synthesis of nanostructures with a developed surface morphology for sensor designs. For example, hillocks developed on the side surface of a relatively thick nanowire can be used as a system of nuclei in the subsequent synthesis of ordered chains of side nanopillars, oriented perpendicular to the nanowire axis. The diffusion regime of deposition of free atoms on the surface of the nanowire is unstable, since the tops of the formed hillocks are zones of high concentration of diffusion fluxes, creating shadow zones around them [61,63,66]. As a result, two or four longitudinal rows of nanopillars (relative to the nanowire axis) can be synthesized on the original nanowires with (110) or (100) orientation, respectively (Figure 1A(b) and Figure 9e).

## Figures and Tables

**Figure 1 nanomaterials-12-01411-f001:**
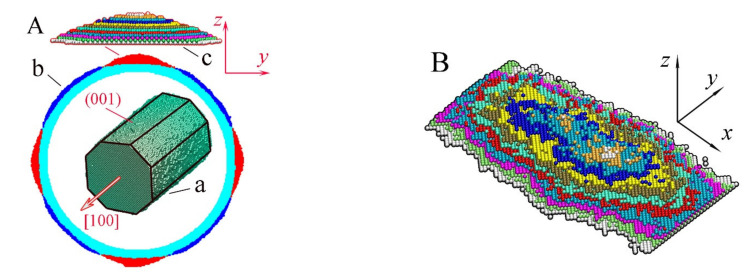
(**A**) Morphology of a thick nanowire with an FCC lattice structure at the initial stage of its evolution: (a) the typical shape of an initially cylindrical nanowire in the absence of the roughening transition effect; (b) nanowire cross-sections with an initial radius of $r_{nw}=55a$ (a —lattice constant) in the area of its maximum size along the y and z axes in two variants of its dynamics for different values of the parameter p at a fixed α=1: the physical meanings of parameters α and p are presented in the description of the model (see Section 2). Atoms marked in cyan and blue represent the near-surface atoms when p=0.7. The cyan and red atoms correspond to the case with p=0.825 (the case of stimulated surface diffusion of atoms). Both cross-sections are calculated at a time moment $t=3\times 10^{6}$ MC time steps; (c) detailed representation of a hillock structure by multi-colored (001) atomic layers; (**B**) a typical hillock having formed on the side nanowire surface by the time moment $t=3\times 10^{6}$ MC time steps. The sizes of the lowest (001) layer (marked in red) are around $l_{x}\sim 53a$, $l_{y}\sim 23a$.

**Figure 2 nanomaterials-12-01411-f002:**
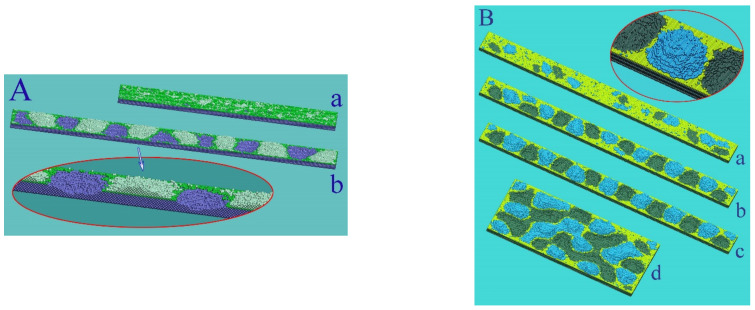
The manifestation of a roughening transition on the upper faces of plates with different orientations. It is assumed that the side faces of the plates, with the exception of the upper face, are in contact with “frozen”/immobile atoms, and the mobile atoms of the plate cannot go beyond the perimeter of the upper face: (**A**) the upper face of the plate is part of the (100) plane. Configuration (a) α=1, p=0.7. The plate length is L=400, width w=70, height h=10. In the quasi-steady state, the atomic layer +1 (the top atomic layer in the initial state is marked with the index 0) contains an approximately constant number of atoms around 1600. The formed surface clusters drift along the zero layer, decay, and reappear. (b) −α=1,p=0.825. L=600, w=70, h=10. $t=2.9\times 10^{6}$ MC steps. The number of broken bonds in the system of mobile atoms, $N_{bb}\left(t\right)$, reaches saturation at $\tau_{sat}\approx 2.5\times 10^{6}$ MC steps. The bottom inset shows the surface structure of the plate in the longitudinal middle section; (**B**) dynamics of a plate with an upper face (111) type. (a–c) L=700, w=70, h=10. α=1, p=0.825. $t=0.5,\;1.0,\;1.5\left(\times 10^{6}\right)\approx \tau_{sat}$. Configuration (d)  L=370, w=200. $t=1.4\times 10^{6}$.

**Figure 3 nanomaterials-12-01411-f003:**
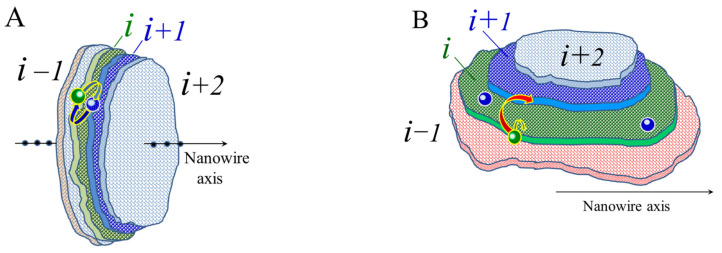
Scheme of a fragment of a modulated nanowire (**A**) and a hillock (**B**) for estimating the calculation of the balance of surface fluxes of atoms from layer i to layer (i+1 ) and back. In subsection (**B**), long jumps marked in red are prohibited.

**Figure 4 nanomaterials-12-01411-f004:**
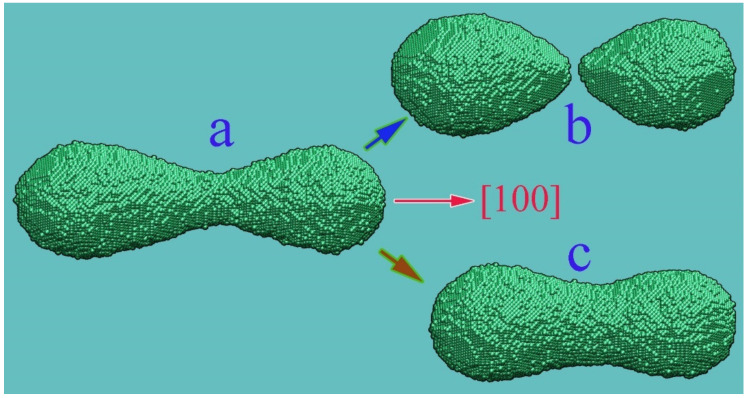
Controlling the direction of surface atomic flows in the nanocluster-narrowing region. Configuration (**b**) is formed from the initial state (**a**) as a result of the outflow of atoms from the narrowing area in the case of a low surface diffusion rate—p=0.7, α=1. When surface diffusion is stimulated, p=0.825,α=1, the influx of atoms into this zone forms the entire cluster (**c**). The presented two-way transformation is stably reproduced in other random realizations of the dynamics of the initial nanocluster.

**Figure 5 nanomaterials-12-01411-f005:**
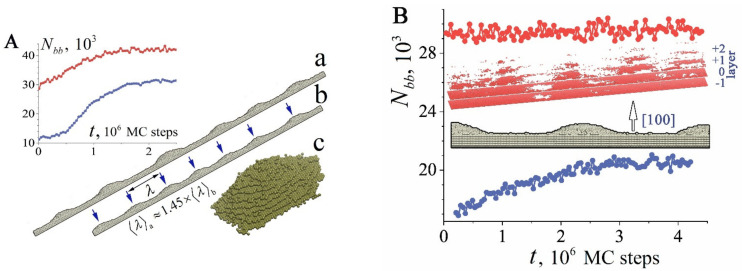
Effect of possible surface heating on the manifestation of stimulated roughening transition on faces of different orientations: (**A**) the outer surface of the plate is a (111)-type facet. The upper inset shows the number of broken bonds (relative to the initial state),$\;N_{bb}\left(t\right)$, in the cold (p=0.825, α=1 —blue curve) and hot (p=0.825, α=0.75 —red curve) regimes. Configurations (a, b) quasi-steady state of the plate surfaces in hot regime with lengths L=700 and 550 (in both cases w=70,h=10 ). The lower inset shows one of the hillocks formed on the upper face of the plate (a); (**B**) results for (100)-orientation of the upper face of the plate. L=550, w=70, h=10. The blue and red circles show time dependencies, $N_{bb}\left(t\right)$, for cold and hot regimes, respectively. In the cold regime, six hillocks with a height of 8–10 atomic layers are formed on the upper plane. The bottom inset presents a fragment of the plate. In the hot regime, only two or three near-surface atomic layers are partially filled (upper inset). The observed low structuredness of the surface is characterized by longer wavelength perturbations.

**Figure 6 nanomaterials-12-01411-f006:**
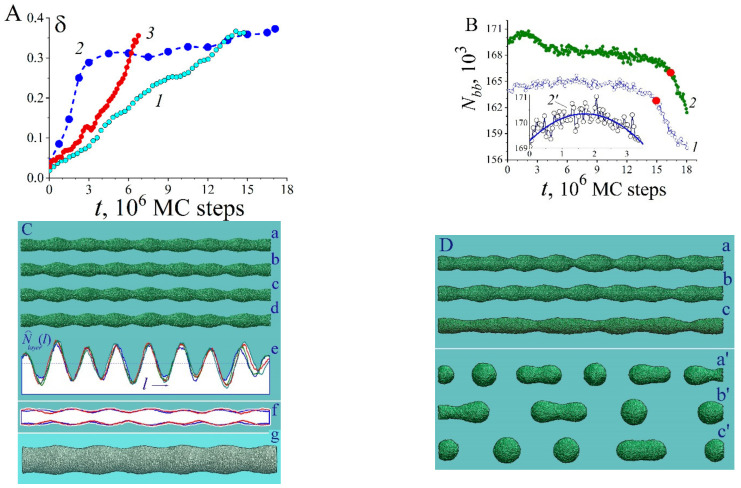
Dynamics of nanowires with (100) orientation in different regimes: (**A**) dependencies of the level of modulation of the cross-section of the nanowire, $\delta \left(t\right)$, on time:$\;\delta =\Delta N_{layer}^{2}/N_{layer}{}^{2}$, $N_{layer}\;$ is a number of atoms in transverse atomic layers. Initial nanowire radius $r_{nw}=15.5$, length L=700. Curve 1 (cyan circles)—p=0.7, α=1 (breakup without external irradiation). Blue circles—p=0.825, α=1 (transformation under external irradiation without heating). The red curve (3)—p=0.825, α=0.8 (breakup under external irradiation with heating). Dependences $\delta \left(t\right)$ are given up to the first nanowire rupture; (**B**) variations in time of the number of broken bonds, $N_{bb}\left(t\right),$ for surface atoms of the nanowire. The parameters for curves 1 and 2 are the same as for curves 1 and 2 presented in Subpart A. The lower inset, curve 2’, represents the initial stage of dependency 2; (**C**) formation of a quasi-equilibrium shape of a nanowire with the pronounced roughening transition. The parameters are the same as shown in subpart (**A**) for the blue curve (2). Configurations (a–d) depict shapes of the nanowire at time moments $t=3,\;6,\;9,\;12(\times 10^{6}$ ) MC steps. For these time moments, subpart (e) shows the distributions of atoms in atomic layers (100) along the nanowire axis by blue, olive, red, dark cyan curves, respectively: ${\hat{N}}_{layer}\left(l\right)=N_{layer}\left(l\right)/N_{layer}\left(t=0\right)$. Subpart (f) represents a longitudinal section of a nanowire with a larger radius ($r_{nw}=20$ ) at time moments $t=9\times 10^{6}$ (blue curve) and $t=36\times 10^{6}$ MC steps (red curve). Configuration (g) depicts the shape of a thick nanowire: $r_{nw}=35$ (p=0.825,  α=1; L=700 ), $t=15.7\times 10^{6}$ MC steps; (**D**) configurations (a–c) depict the nanowire shapes close to the first breakup for regimes (1), (2), and (3) presented in subpart A; $t=13.5,\;12,\;6.5(\times 10^{6}$ ) MC steps. For these cases parameter δ≈0.33. Configurations (a’–c’) show the nanowire shapes at the final stages of breakup: $t=27.9,\;27.9,\;11.7\;\left(\times 10^{6}\right)$ MC steps, respectively.

**Figure 7 nanomaterials-12-01411-f007:**
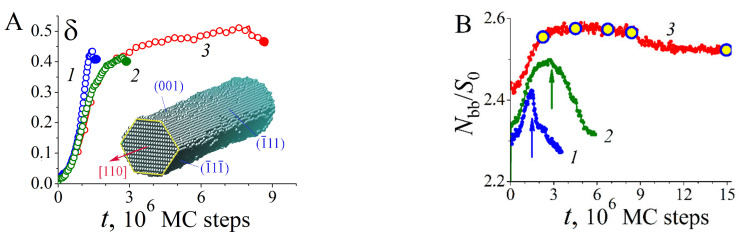
Dynamics of morphology characteristics of nanowires with (110) orientation under stimulation of atomic surface diffusion. α=1, p=0.825: (**A**) change in the level of modulation of the cross-section of nanowires, $\left(t\right)$, of different radii. For curves 1, 2, 3—$r_{nw}=15.5$, L=700; $r_{nw}=17.5$, L=700; and $r_{nw}=20.5$, L=500, respectively. Color circles mark the time moments of the first ruptures. The inset shows the typical shape of a nanowire with (110) orientation established at the initial stage of its transformation; (**B**) change in time of the number of broken bonds on the surface of nanowires for the same parameters. $S_{0}$ is the side surface area of the nanowire in dimensionless units in the initial state. The arrows indicate the moments of the first ruptures in the nanowires. The circles on curve 3 mark the moments of time for which in Figure 8C the shapes of the thickest nanowire are shown.

**Figure 8 nanomaterials-12-01411-f008:**
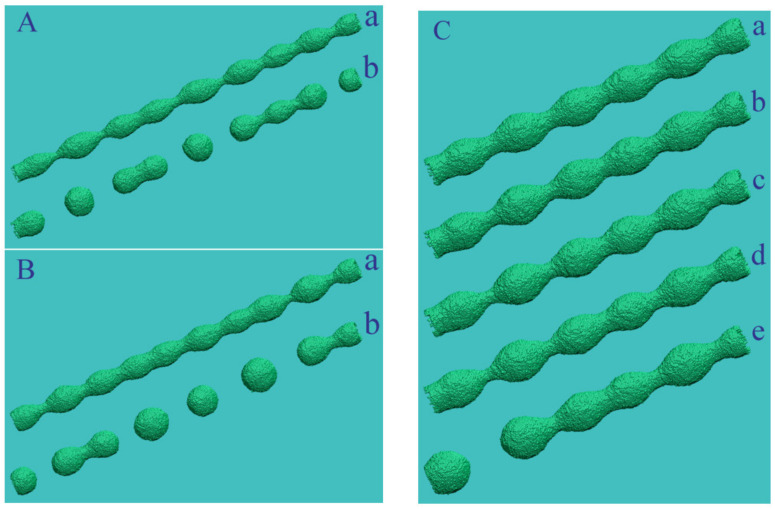
Shapes of nanowires with (110) orientation at different stages of transformation upon stimulation of surface diffusion of atoms: α=1, p=0.825: (**A**) $r_{nw}=15.5$, L=700; for configurations (a,b) t=1.50 and $3.54\left(\times 10^{6}\right)$ MC steps; (**B**) $r_{nw}=17.5$, L=700; (a,b) t=2.70 and $5.88\left(\times 10^{6}\right)$ MC steps. (**C**) $\;r_{nw}=20.5$, L=500; (a–e) t=2.25, 4.50, 6.75,8.40 and 15.0$\left(\times 10^{6}\right)$ MC steps (these points in time are marked with circles on the dependence $N_{bb}\left(t\right)$ —see curve 3 in Figure 7B).

**Figure 9 nanomaterials-12-01411-f009:**
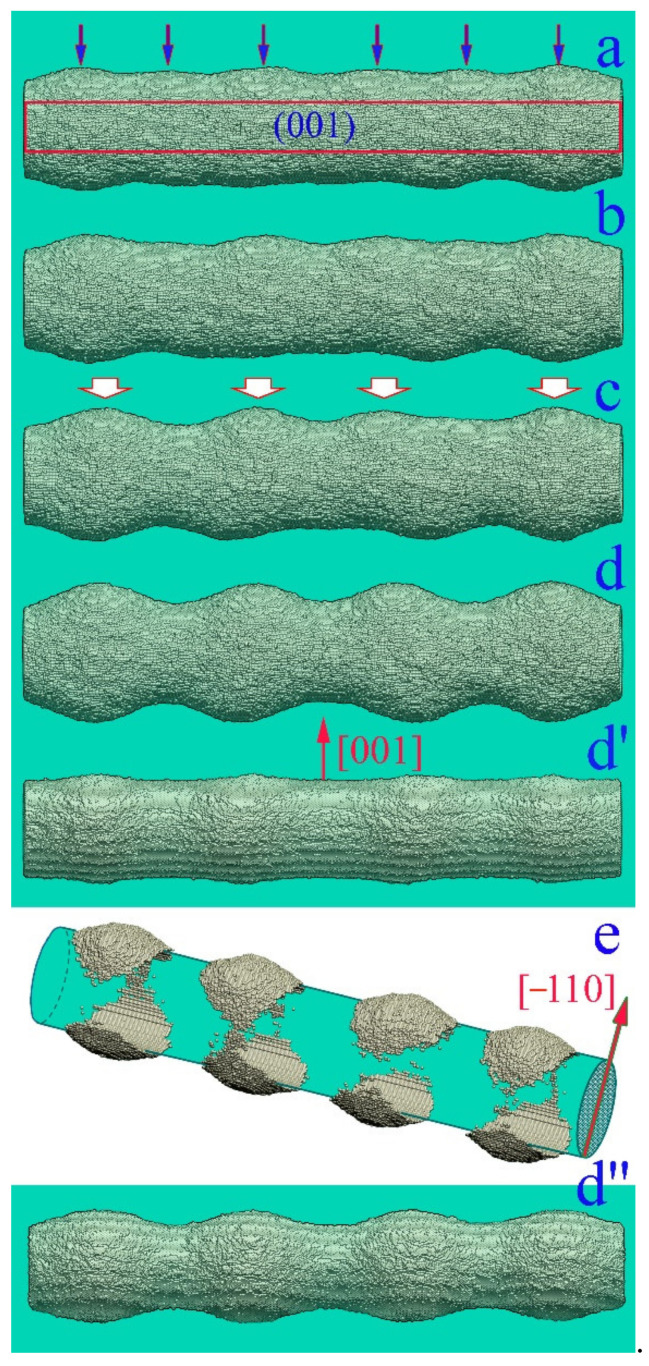
Surface dynamics of a nanowire with radius $\;r_{nw}=45$ ($\;r_{nw}\approx 18\;nm$ for Au) under conditions of stimulated surface diffusion of atoms ((110) orientation, L=500 ). Subparts (a-d) show the shape of the nanowire at times $t=2.25;3.00;3.90;5.10\left(\times 10^{6}\right)$ MC steps (top view along direction (00−1)). Configuration (e) displays the atoms that have gone beyond the initial cylindrical boundary of the wire (green cylinder) for configuration (d’). Configurations (d’) and (d’’) show the shape of the wire (d) from the side—when viewed along the plane (001)—at time moments $t=5.10\times 10^{6}$ and $t=12.6\times 10^{6}$ MC steps.

**Figure 10 nanomaterials-12-01411-f010:**
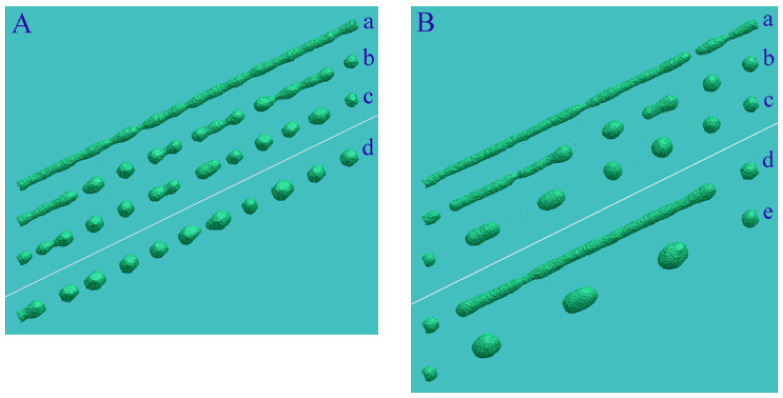
Breakup of nanowires with (111) orientation without external irradiation (subpart (**A**)) and under conditions of stimulated surface diffusion of atoms (subpart (**B**)): (**A**) α=0.9, p=0.725, L=1050. Configurations (a–c) $r_{nw}=12.5$; $t=3.0,\;4.2,\;5.4\left(\times 10^{6}\right)\;$ MC steps. Configuration (d) $r_{nw}=14.5$; $t=9.45\left(\times 10^{6}\right)$ MC steps; (**B**) α=0.9, p=0.8, L=1050. Configurations (a–c) - $r_{nw}=12.5$; $t=3.0,\;4.8,\;7.2\left(\times 10^{6}\right)\;$ MC steps. Configurations (d,e) $r_{nw}=14.5$; $t=12.0,\;26.1\left(\times 10^{6}\right)\;$ MC steps.

## Data Availability

The data presented in this study are available on request from the corresponding author. The data are not publicly available due to a large amount of information.

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
