# Peer review of "Dynamics of Quasi-One-Dimensional Structures under Roughening Transition Stimulated by External Irradiation"

_nanomaterials, 2022, doi:10.3390/nano12091411_

Round 1
Reviewer 1 Report
In this work, the authors studied the dynamics of the nanowire surfaces upon various modulations. In particular, they emphasized the importance of the rough transition phenomenon, as opposed to the widely discussed Rayleigh instability, in certain circumstances like surface irradiation. Monte Carlo (MC) model was employed to conduct the numerical experiments of different scenarios. Two parameters, p and α were introduced in the model to simulate different conditions, and they found a striking result that after reaching a certain value of p (via surface stimulation), the direction of the structural evolution would be reversed from breaking up to fusing together. Next, transformation of nanowires with different orientations (to the crystal lattice) were studied in detail, as the rough transition was found occurring faster on (111) than on (100) and (110) facets. Transformation modes that controlled by either Rayleigh instability or rough transition, and both two factors were described.
Overall, the manuscript gave a comprehensive theoretical study on the surface diffusion and the resulting morphological evolution of nanowires. The work can be beneficial for both theoretical development and experimental design and synthesis. It may be published after addressing the following concerns.
- While the authors kept on referring to the experimental result of “external irradiation”, and use it as the verification of the numerical experiments, the in-text cited Ref. [32] and [58] are not the respective works about the Au and Ag nanowires.
- The distinctness between the two mechanisms, Rayleigh instability and the rough transition, needs to be better articulated. How could Rayleigh instability effect be avoided when there is heat involved in the irradiation process? The long-wave transformation route does not involve the rough transition because of the (110) facets. Does that mean the surface stimulation also contribute the Rayleigh instability?
- The presentation of the figures needs to be improved. Some captions contains wired symbols.
- Although external irradiation might be the starting point of this theoretical study, without experimental demonstration, it seems that a more general term, such as “surface stimulation” might be a more appropriate description to the calculated scenario.
- The reference list needs to be thoroughly checked, as almost none of them are correctly linked to the in-text citation. The language need to be revised to improve the logic and readability. Font and number notations needs to be consistent.
Reviewer 2 Report
The paper entitled “Dynamics of quasi-one-dimensional structures under roughening transition stimulated by external irradiation” by Gorshkov et al, deals with a Monte Carlo simulation of the effect of external irradiation of nanowires on the dynamics of their surface morphology at elevated temperatures while preserving their crystal lattice with the possibilities of controllable periodic modulation of the cross-section of quasi-one-dimensional structures.
The paper is well presented, and the results are adequately Analysed with the supporting simulation data. I have only one concern about this publication which is directly related to a very similar paper published by the same authors last year (DOI: 10.1039/D0CE01404D).
The paper has been cited in this manuscript, however, the authors need to add a paragraph where they explicitly discuss their former results and show the novelty introduced in the current paper
Round 2
Reviewer 2 Report
The revised paper can now be accepted for publication